# INCONSISTENCY BIASES IN DYNAMIC DATA PRUNING

**Qing Zhou**[1]  **Tao Yang**[1]  **Bingxuan Zhao**[1]  **Hongyuan Zhang**[2]  **Junyu Gao**[1*]  **Qi Wang**[1*]
[1]Northwestern Polytechnical University    [2]The University of Hong Kong
{chautsing, ytao9464, bxuanzhao202, hyzhang98, gjy3035, crabwq}@gmail.com

## ABSTRACT

Dynamic data pruning accelerates training by focusing on informative samples. However, comparing importance scores across different model states introduces inconsistency (score context drift), and variable selection rates bias gradient dynamics over time (temporal gradient bias). We introduce RePB (Resolving Pruning Biases), a framework addressing these issues. RePB performs pruning decisions within local windows (short sequences of batches) during training, using loss scores computed with a near-constant model state within each window to ensure valid comparisons. These decisions determine the data subset used in the subsequent training phase. To counteract temporal gradient bias arising from non-uniform sample inclusion, cumulative temporal rescaling reweights sample losses during training based on their historical selection frequency. We provide theoretical grounding for RePB's consistency in score comparison and gradient alignment. Experiments show RePB achieves near-full-dataset accuracy using reduced data (most above 30%) across 16 datasets, 17 models and 13 tasks, offering a robust and scalable approach to efficient deep learning. Code is available at https://github.com/mrazhou/RePB.

## 1 INTRODUCTION

The remarkable success of deep learning models across diverse applications (He et al., 2016; Dosovitskiy et al., 2021a) often comes at the cost of immense computational resources and vast datasets (Kaplan et al., 2020; Russakovsky et al., 2015; Feng et al., 2019). As models continue to grow in complexity and datasets expand, improving training efficiency without compromising model quality becomes paramount. Data selection, which aims to train models on smaller, carefully chosen data subsets, represent a promising direction for achieving such efficiency gains (Coleman et al., 2020; Raju et al., 2021; Sener & Savarese, 2018; He et al., 2024).

Among data selection techniques, dynamic methods that adapt the training subset during the learning process hold particular appeal (Qin et al., 2024; Zhou et al., 2025; Raju et al., 2021). By responding to the changing importance or redundancy of data points as the model evolves, dynamic pruning can potentially offer greater efficiency than static selection fixed before training (Coleman et al., 2020; Paul et al., 2021; Toneva et al., 2019). However, the practical effectiveness and reliability of dynamic pruning are often undermined by two fundamen-

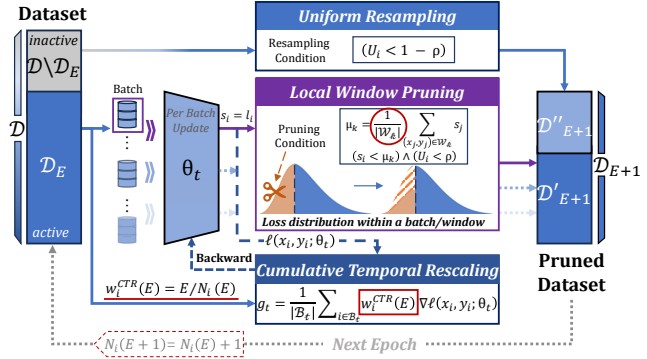

Figure 1: Overview of RePB framework.

tal consistency issues. First, **score context drift**: importance metrics (e.g., sample loss Qin et al. (2024); Zhou et al. (2025); Raju et al. (2021), gradient norm Katharopoulos & Fleuret (2018)) used for pruning decisions are typically evaluated using the current model state. Because the model parameters (the context) drift significantly during training, comparing scores computed at different

---

[*]Co-corresponding author

steps lacks statistical validity and can lead to suboptimal pruning choices (Blalock et al., 2020). Second, **temporal gradient bias**: iteratively selecting non-uniform subsets epoch after epoch alters the effective sampling distribution over time compared to standard uniform sampling from the full dataset. This introduces a bias in the expected cumulative gradient trajectory, potentially hindering convergence or leading the model to different optima (Qin et al., 2024; Hu et al., 2020). These consistency violations represent fundamental obstacles to achieving reliable and theoretically sound acceleration with dynamic data pruning.

To overcome these critical limitations, particularly *the inconsistency biases inherent in scoring context and temporal gradient dynamics*, we propose RePB (Resolving Pruning Biases), a framework characterized by *pruning within a batch* and *rescaling over epochs* (As shown in the Figure 1). Specifically, to ensure reliable pruning decisions despite score context drift, it employs *local window pruning (LWP)*: importance scores (sample losses) used to determine the data for the next epoch are computed and compared only within short local windows of the current epoch (e.g., one or a few batches). Leveraging the minimal model drift within these short windows allows for valid comparisons of scores generated at slightly different steps, enabling reliable identification of less informative samples relative to their peers within the same window. Complementing this selection process, RePB maintains data diversity and prevents sample pool collapse through *uniform probability resampling*. At the end of each epoch, it explicitly reintroduces samples from the original dataset that were not part of the current training set with a fixed probability, guaranteeing long-term exploration. Finally, to counteract the temporal bias in gradient dynamics arising from selecting potentially non-uniform subsets over epochs, RePB utilizes *cumulative temporal rescaling (CTR)*. During training in the subsequent epoch, CTR reweights the loss of each sample based on its historical selection frequency, effectively applying an empirical inverse probability weighting that aims to align the expected gradient trajectory with that of full-dataset training over time. By systematically addressing both score comparison validity and long-term gradient bias, this synergistic approach allows RePB to deliver the efficiency benefits of dynamic pruning without sacrificing the stability and reliability of standard training.

## 2 RELATED WORK

**Data Selection and Pruning.** Selecting informative data subsets is a core theme in machine learning, pursued for efficiency (Konyushkova et al., 2017; Kousar et al., 2025) and generalization (Jiang et al., 2018). Static subset selection methods choose data once before training. Early static methods were often based on criteria like core-sets (Huggins et al., 2016), uncertainty (Coleman et al., 2020), or gradient information (Katharopoulos & Fleuret, 2018; Paul et al., 2021). More recent work in static pruning has developed sophisticated criteria that move beyond just efficiency, focusing on enhancing model generalization and data quality. For instance, methods have been proposed to improve generalization by severing spurious correlations (Mulchandani & Kim, 2025) or by carefully selecting data in specialized domains like molecular modeling (Chen et al., 2024). Other advanced static criteria include perplexity-based scoring with small reference models (Ankner et al., 2025), information maximization (Tan et al., 2025), and influence-based metrics like moving-one-sample-out (Tan et al., 2023). While these static approaches are powerful, they select data once and cannot adapt to the model's evolving understanding of the dataset during training. In contrast, dynamic methods, like InfoBatch, update the subset during training (Toneva et al., 2019; Qin et al., 2024). Early work often focused on identifying forgettable samples (Toneva et al., 2019), while more recent approaches use loss dynamics (Qin et al., 2024; Zhou et al., 2025). RePB falls into the dynamic category, but it distinguishes itself by addressing the fundamental optimization consistency biases inherent in the dynamic process itself, rather than focusing solely on static data quality.

**Addressing Score Inconsistency and Staleness.** The challenge of comparing importance scores computed across differing model states is recognized in the literature (Jiang et al., 2019; Borsos et al., 2020). Prior attempts to mitigate this include using score moving averages (Jiang et al., 2019) or employing infrequent score updates (Borsos et al., 2020). However, these strategies often provide indirect mitigation rather than a direct structural guarantee of comparable scoring contexts. RePB's local window pruning offers a more *direct and foundational solution* by explicitly restricting score comparisons to windows of model state stability, thereby ensuring local decision consistency and tackling the root cause of this specific inconsistency.

**Addressing Gradient Bias and Variance.** Dynamic subset selection inevitably leads to non-uniform sampling over time, potentially biasing gradient estimates (Qin et al., 2024). Importance Sampling (IS) is a classical technique used in stochastic optimization to reduce variance or correct bias by weighting gradient steps inversely to sampling probabilities (Needell et al., 2016; Katharopoulos & Fleuret, 2018; Qin et al., 2024; Salaün et al., 2023; 2024). IS often uses instantaneous importance or sampling probabilities to adjust the current gradient step, primarily for variance reduction. CTR, conversely, uses cumulative selection frequencies (counts of inclusion in past epochs' datasets) to correct the long-term temporal bias introduced by the selection process itself across epochs. It aims to align the entire training trajectory, not just individual steps (detailed in Appendix A).

In conclusion, RePB offers a refined approach to dynamic data pruning by addressing critical consistency challenges. While methods like InfoBatch (Qin et al., 2024) tackle temporal gradient bias through instantaneous rescaling, RePB introduces a distinct cumulative temporal rescaling mechanism. This approach ensures long-term gradient alignment with enhanced applicability, as it operates effectively without relying on explicit knowledge or modeling of precise selection probability distributions, which may be intractable or unavailable. Crucially, RePB also identifies and resolves the challenge of score context drift—a limitation particularly evident in global comparison strategies like InfoBatch's—through its local window scoring strategy, which ensures valid importance comparisons within stable model contexts. RePB thus provides a more robust, theoretically grounded, and broadly applicable path to efficient training than prior dynamic pruning methods.

## 3 RePB: Resolving Inconsistency Biases in Dynamic Pruning

Dynamic data pruning seeks efficiency gains by training on subsets of data identified as most informative. However, two fundamental consistency challenges hinder its reliability: (1) Score context drift: Importance scores (e.g., sample losses) computed at different training stages (with different model parameters, i.e., different contexts) are not directly comparable, making pruning decisions unreliable. (2) Temporal gradient bias: Selecting subsets non-uniformly over time alters the expected cumulative gradient compared to full-dataset training. RePB (Resolving Pruning Biases, as shown Figure 1) tackles these issues systematically using local window pruning, uniform probability resampling, and Cumulative Temporal Rescaling (CTR).

### 3.1 Local Window Pruning for Selection Consistency

During the training process on dataset $\mathcal{D}_E$ (in epoch $E$), RePB identifies samples within $\mathcal{D}_E$ for potential inclusion in $\mathcal{D}_{E+1}$. Pruning decisions are localized within windows. A window $k$ may consist of a single batch $\mathcal{B}_t$ or span $W$ consecutive batches (e.g., from step $t_k$ to $t_{k+W-1}$); the latter is particularly useful with small training batch sizes to ensure a sufficient sample pool for effective pruning. We denote the set of samples processed in window $k$ as $\mathcal{W}_k \subseteq \mathcal{D}_E$. For each sample $(x_i, y_i) \in \mathcal{W}_k$ processed at step $t \in [t_k, t_{k+W-1}]$, its importance score is computed using the model state at that time: $s_i = \ell(x_i, y_i; \theta_t)$. After processing all samples in the window, these scores $\{s_i\}_{(x_i, y_i) \in \mathcal{W}_k}$ are collected. The core assumption is that the parameter drift $\|\theta_t - \theta_{t'}\|$ for any $t, t'$ within the window $[t_k, t_{k+W-1}]$ is small. This allows for a meaningful comparison of the scores $s_i$ collected across the window.

Within each processing window $\mathcal{W}_k$, importance scores $s_i = \ell(x_i, y_i, \theta_{\text{window}})$ are computed for all constituent samples $(x_i, y_i) \in \mathcal{W}_k$. The window-specific mean score is then determined as $\mu_k = \frac{1}{|\mathcal{W}_k|} \sum_{(x_j, y_j) \in \mathcal{W}_k} s_j$. Let $\rho \in [0, 1]$ be a pre-defined hyperparameter representing the probability with which a sample $(x_i, y_i) \in \mathcal{W}_k$ having $s_i < \mu_k$ is pruned. For each sample $(x_i, y_i) \in \mathcal{W}_k$, an independent random variable $U_i \sim \mathcal{U}(0, 1)$ is drawn. The sample is retained and included in the candidate set $\mathcal{D}'_{E+1}$ for the subsequent epoch $E + 1$ if the following condition holds:

$$(s_i \geq \mu_k) \vee (s_i < \mu_k \wedge U_i \geq \rho) \tag{1}$$

Otherwise, the sample is temporarily pruned. This selection rule is applied to all samples across all windows processed during epoch $E$, and the union of retained samples forms $\mathcal{D}'_{E+1}$.

## 3.2 UNIFORM PROBABILITY RESAMPLING FOR DIVERSITY

After processing $\mathcal{D}_E$ and collecting retained samples into $\mathcal{D}'_{E+1}$ at the end of epoch $E$, resampling occurs to ensure diversity and counteract sample pool shrinkage. We identify the set of samples from the full dataset $\mathcal{D}$ that were not part of the training set for the current epoch $E$, i.e., $\mathcal{D} \setminus \mathcal{D}_E$. Every sample $(x_j, y_j) \in \mathcal{D} \setminus \mathcal{D}_E$ is added to form the final dataset $\mathcal{D}_{E+1}$ with probability $\rho_{\text{resample}} = 1 - \rho$.

$$\text{Final } \mathcal{D}_{E+1} = \mathcal{D}'_{E+1} \cup \{(x_j, y_j) \in \mathcal{D} \setminus \mathcal{D}_E \mid \text{random}(0, 1) < \rho_{\text{resample}}\}. \tag{2}$$

This step guarantees that samples outside the current training focus have a chance to re-enter, preventing convergence to a empty subset.

## 3.3 CUMULATIVE TEMPORAL RESCALING FOR GRADIENT ALIGNMENT

Dynamic selection can cause samples to be used with varying frequencies over epochs, deviating from the uniform usage in standard training and potentially biasing the cumulative gradient. CTR counteracts this during the training phase of epoch $E + 1$ by amplifying the gradient contribution of under-selected samples and dampening that of over-selected ones.

Let $N_i(E)$ be the total number of times sample $(x_i, y_i)$ has been included in the training datasets from epoch 1 to $E$:

$$N_i(E) = \sum_{e=1}^{E} \mathbb{1}[(x_i, y_i) \in \mathcal{D}_e], \tag{3}$$

where $\mathbb{1}[\cdot]$ is the indicator function. Assuming the initial dataset $\mathcal{D}_1$ contains all samples from the full dataset $\mathcal{D}$, every sample $i$ has $N_i(E) \geq 1$ for $E \geq 1$.

The core idea of CTR is to reweight each sample's contribution inversely proportional to its observed historical selection frequency relative to the baseline (being selected in every epoch). We define the CTR weight for sample $i$ to be used during epoch $E + 1$ as: $w_i^{\text{CTR}}(E) = E/N_i(E)$. This weight $w_i^{\text{CTR}}(E)$ represents the ratio of the number of epochs elapsed to the number of times sample $i$ was actually used. If $N_i(E) < E$, the sample was under-selected, and its weight will be greater than 1. If $N_i(E) > E$ (possible if resampling allows multiple inclusions), it was over-selected, and its weight is less than 1. If $N_i(E) = E$, its weight is 1, matching standard training.

During training in epoch $E + 1$, when processing a mini-batch $\mathcal{B}_t \subset \mathcal{D}_{E+1}$, the gradient update is computed using these weights directly:

$$\mathbf{g}_t = \frac{1}{|\mathcal{B}_t|} \sum_{i \in \mathcal{B}_t} w_i^{\text{CTR}}(E) \nabla \ell(x_i, y_i; \theta_t). \tag{4}$$

The standard mini-batch averaging $1/|\mathcal{B}_t|$ is retained, but each sample's gradient $\nabla \ell_i$ is scaled by its individual historical weight $w_i^{\text{CTR}}(E)$. This approach directly uses the historical counts $N_i(E)$ to adjust gradient magnitudes, offering a practical mechanism to mitigate temporal selection bias without requiring knowledge of the underlying selection probabilities.

## 4 THEORETICAL FOUNDATIONS

### 4.1 LOCAL WINDOW PRUNING ENSURES SCORE CONSISTENCY

**Proposition**: By restricting pruning decisions to comparisons of scores collected within local windows, RePB leverages the bounded and typically small parameter drift within such windows. This makes the comparison of scores computed at slightly different model states significantly more reliable than comparisons across larger time intervals (e.g., epochs), thus mitigating the impact of score context drift.

**Proof**: Consider the $k$-th window $\mathcal{W}_k$ spanning steps $t_k$ to $t_{k+W-1}$. For a sample $(x_i, y_i) \in \mathcal{W}_k$ processed at step $t \in [t_k, t_{k+W-1}]$, its score is $s_i = \ell(x_i, y_i; \theta_t)$. A comparison might involve $s_i$ and $s_j = \ell(x_j, y_j; \theta_{t'})$ where $t, t' \in [t_k, t_{k+W-1}]$, or comparing $s_i$ to the window mean $\mu_k = \frac{1}{|\mathcal{W}_k|} \sum_{j \in \mathcal{W}_k} \ell(x_j, y_j; \theta_{t'_j})$.

The validity of such comparisons relies on the parameter drift within the window being small. Assume the loss function $\ell(x, y; \theta)$ is $L$-Lipschitz continuous with respect to parameters $\theta$. If gradient norms are bounded by $G$ and learning rate is $\eta$, the parameter change between any two steps $t, t'$ within the window is bounded: $\|\theta_t - \theta_{t'}\| \leq |t - t'|\eta G \leq W\eta G$. The difference in loss for the same sample $i$ evaluated at two different model states $\theta_t, \theta_{t'}$ within the window is bounded:

$$|\ell(x_i, y_i; \theta_t) - \ell(x_i, y_i; \theta_{t'})| \leq L\|\theta_t - \theta_{t'}\| \leq LW\eta G. \tag{5}$$

If the window size $W$ and learning rate $\eta$ are sufficiently small, this difference is small. This implies that the score $s_i = \ell(x_i, y_i; \theta_t)$ is a reasonably stable measure of the sample's importance relative to the model states encountered within that window. Comparing $s_i$ to $s_j$ (computed with $\theta_{t'}$) or to the mean $\mu_k$ (an average over slightly different $\theta$'s) is meaningful because the score context drift within the window is limited. The relative ordering of sample importance is likely preserved. The efficacy of LWP relies on minimal intra-window parameter drift. This is perfectly achieved when a window is a single batch ($W = 1$), as scores are computed with a fixed model state before any parameter update, resulting in zero drift. Since extremely small batch sizes are uncommon in many settings, $W = 1$ often serves as an ideal and practical default for RePB. For windows spanning multiple batches ($W > 1$), the anticipated drift (proportional to $W\eta G$) remains **significantly smaller than cumulative drift over an entire epoch**. Thus, unlike epoch-wide score aggregation which suffers from substantial score context drift, LWP ensures a much higher degree of score comparability, providing a more statistically sound basis for selection.

## 4.2 Cumulative Temporal Rescaling Reduces Gradient Bias

**Proposition**: By reweighting gradient contributions during training using weights $w_i^{\text{CTR}}(E) = E/N_i(E)$ derived from empirically observed historical selection counts, CTR ensures that the expectation of the stochastic gradient updates aligns with the direction of the gradient computed over the full dataset $\mathcal{D}$, correcting the bias induced by non-uniform dynamic selection frequencies.

**Proof**: The target unbiased gradient over the full dataset $\mathcal{D}$ is $\mathbf{g}^*(\theta) = \frac{1}{|\mathcal{D}|} \sum_{i \in \mathcal{D}} \nabla \ell_i(\theta)$. The CTR method computes the stochastic gradient at step $t$ (within epoch $E + 1$) using the weight $w_i^{\text{CTR}}(E) = E/N_i(E)$ based on history up to epoch $E$:

$$\mathbf{g}_t = \frac{1}{|\mathcal{B}_t|} \sum_{i \in \mathcal{B}_t} w_i^{\text{CTR}}(E) \nabla \ell_i(\theta_t), \tag{6}$$

where $\mathcal{B}_t$ is sampled uniformly from the current epoch's dataset $\mathcal{D}_{E+1}$. We analyze the expectation of this gradient $\mathbb{E}[\mathbf{g}_t \mid \theta_t]$. Taking expectations over the batch sampling and the dataset selection process, and assuming $|\mathcal{D}_{E+1}| \approx S_{E+1}$, we have:

$$\mathbb{E}[\mathbf{g}_t \mid \theta_t] \approx \frac{1}{S_{E+1}} \sum_{i \in \mathcal{D}} p_{i,E+1} \cdot \mathbb{E}[w_i^{\text{CTR}}(E)] \cdot \nabla \ell_i(\theta_t). \tag{7}$$

The weight $w_i^{\text{CTR}}(E) = E/N_i(E)$ is the reciprocal of the empirical selection frequency $f_i(E) = N_i(E)/E$. By the Law of Large Numbers, $f_i(E) \to \bar{p}_i$ as $E \to \infty$, where $\bar{p}_i = \lim_{E \to \infty} \frac{1}{E} \sum_{e=1}^{E} p_{i,e}$ is the long-term average selection probability of sample $i$. This makes $w_i^{\text{CTR}}(E)$ a practical, computable estimator for $1/\bar{p}_i$.

Here, we use the approximation $\mathbb{E}[w_i^{\text{CTR}}(E)] = \mathbb{E}[1/f_i(E)] \approx 1/\mathbb{E}[f_i(E)] \approx 1/\bar{p}_i$. As noted by Jensen's inequality, that $\mathbb{E}[1/f_i(E)] \geq 1/\mathbb{E}[f_i(E)]$. This approximation becomes increasingly accurate as $E \to \infty$, because the Strong Law of Large Numbers ensures that the random variable $f_i(E)$ concentrates tightly around its mean $\bar{p}_i$, reducing the error of the approximation.*

Substituting this approximation into Eq. 7:

$$\mathbb{E}[\mathbf{g}_t \mid \theta_t] \approx \frac{1}{S_{E+1}} \sum_{i \in \mathcal{D}} p_{i,E+1} \frac{1}{\bar{p}_i} \nabla \ell_i(\theta_t). \tag{8}$$

---

*Practically, the inequality implies that our estimator may slightly overweight samples, especially those with low or highly variable selection counts. This can be viewed as a beneficial conservative correction, as it gives a slightly larger voice to under-selected samples, helping to prevent catastrophic forgetting and aiding robust convergence.

As the selection process stabilizes, the current selection probability $p_{i,E+1}$ will fluctuate around the long-term average $\bar{p}_i$. Assuming $p_{i,E+1} \approx \bar{p}_i$, the ratio $p_{i,E+1}/\bar{p}_i \approx 1$. Then:

$$\mathbb{E}[\mathbf{g}_t \mid \theta_t] \approx \frac{1}{S_{E+1}} \sum_{i \in \mathcal{D}} \nabla \ell_i(\theta_t) = \frac{|\mathcal{D}|}{S_{E+1}} \mathbf{g}^*(\theta_t). \tag{9}$$

This demonstrates that the expected gradient under CTR, $\mathbb{E}[\mathbf{g}_t]$, is approximately proportional to the true unbiased gradient $\mathbf{g}^*(\theta_t)$. The proportionality constant is $C' = |\mathcal{D}|/S_{E+1}$. If the selection strategy maintains a roughly constant expected subset size fraction $\alpha = S_{E+1}/|\mathcal{D}|$, then $C' \approx 1/\alpha$, a stable factor.

This factor $C'$ acts as a global scaling of the gradient. The outcome of CTR is the correction of the *relative* contributions of different samples in the expected gradient, aligning the optimization direction with that of full-dataset training. The resampling mechanism helps ensure $N_i(E)$ grows over time for all $i$, keeping the weights well-behaved and ensuring the concentration of $f_i(E)$.

## 5 EXPERIMENTS

### 5.1 EXPERIMENTAL SETUP

**Datasets and Tasks.** Our experiments span a multitude of learning paradigms and data modalities. For image classification, we use CIFAR10, CIFAR100 (Krizhevsky et al.), and ImageNet-1K (Russakovsky et al., 2015). Vision-language understanding is evaluated on zero-shot captioning with ToCa (Zhou et al., 2024), cross-domain captioning with SS1M (Feng et al., 2019), image captioning on COCO (Chen et al., 2015), and video captioning on MSR-VTT (Xu et al., 2016). Scene text recognition experiments utilize the MJ+ST dataset (Jaderberg et al., 2014; Gupta et al., 2016). We further test RePB on multi-view stereo with WHU-MVS (Liu & Ji, 2020), cross-view geolocalization with CVACT (Liu & Li, 2019), and image generation on MNIST (Deng, 2012) and CIFAR10. Finally, we explore supervised and semi-supervised learning for image (EuroSAT (Helber et al., 2019), CIFAR100), text (AG News (Zhang et al., 2015), Yelp Review (Yelp, Inc.)), and audio (ESC-50 (Piczak, 2015)).

**Models.** We demonstrate RePB's architecture independence by applying it to Convolutional Neural Networks (CNNs: ResNet18, ResNet50 (He et al., 2016), EfficientNet (Tan & Le, 2019)), Transformers (ViT (Dosovitskiy et al., 2021b), Swin Transformer (Liu et al., 2021), ViECap (Fei et al., 2023), ABINet (Fang et al., 2021), BERT (Devlin et al., 2019), HuBERT (Hsu et al., 2021), GeoDTR (Zhang et al., 2023)), State Space Models (Mamba-based Vim (Zhu et al., 2024)), and other task-specific architectures like Ada-MVS (Liu et al., 2023), VAE Kingma & Welling (2014); Ho & Salimans (2022), DDPM Ho et al. (2020), FixMatch (Sohn et al., 2020), FlexMatch (Zhang et al., 2021), and Dash (Xu et al., 2021).

**Efficiency Metric: Pruning Rate.** We use the *pruning rate* (or Pruned %), defined as the percentage of data points skipped during training, as our primary indicator of efficiency. RePB's computational overhead for score calculation (reusing existing sample losses) and rescaling is minimal and largely amortized. Therefore, the pruning rate serves as a hardware-independent, and easily comparable metric that directly reflects the reduction in computational load (e.g., forward/backward passes) and, consequently, potential training time speedup. Higher pruning rates, while maintaining or improving task performance, indicate greater efficiency (detailed justification in B)

### 5.2 COMPARATIVE ANALYSIS WITH STATE-OF-THE-ART METHODS

**Dynamic Data Pruning Benchmarks.** Table 1 presents results on CIFAR10 and CIFAR100 using ResNet18. RePB consistently outperforms other static or dynamic pruning methods (e.g., GraNd-4, EL2N-20, DP, Random*, UCB), across all pruning rates. Notably, RePB achieves performance comparable to or even slightly exceeding full dataset training at 30% and 50% pruning rates. Compared to InfoBatch (when run for the same number of epochs, denoted by [‡]), RePB demonstrates superior accuracy, especially at higher pruning rates (e.g., 50% on CIFAR100: 78.1% for RePB vs. 77.7% for InfoBatch[‡]). Even when InfoBatch uses adjusted (typically more) epochs ([†]), RePB often matches or surpasses its performance with less data (e.g., CIFAR10 70%: RePB 94.9% vs.

Table 1: Performance comparison of RePB with state-of-the-art dynamic data pruning methods on CIFAR10 and CIFAR100 using ResNet18 across various pruning rates. $\dagger$: Adjusted epochs as reported in the original paper. $\ddagger$: Same number of epochs as other compared methods. $\backslash$: Indicates an unattainable pruning rate. Random* indicates dynamic random pruning Qin et al. (2024).

| Method | CIFAR10 | | | CIFAR100 | | |
|---|---|---|---|---|---|---|
| | 30% | 50% | 70% | 30% | 50% | 70% |
| ResNet18 | 95.6 | | | 78.2 | | |
| Random | 94.6 $\downarrow_{1.0}$ | 93.3 $\downarrow_{2.3}$ | 90.2 $\downarrow_{5.4}$ | 73.8 $\downarrow_{4.4}$ | 72.1 $\downarrow_{6.1}$ | 69.7 $\downarrow_{8.5}$ |
| GraNd-4 (Paul et al., 2021) | 95.3 $\downarrow_{0.3}$ | 94.6 $\downarrow_{1.0}$ | 91.2 $\downarrow_{4.4}$ | 74.6 $\downarrow_{3.6}$ | 71.4 $\downarrow_{6.8}$ | 68.8 $\downarrow_{9.4}$ |
| EL2N-20 (Toneva et al., 2019) | 95.3 $\downarrow_{0.3}$ | 95.1 $\downarrow_{0.5}$ | 91.9 $\downarrow_{3.7}$ | 77.2 $\downarrow_{1.0}$ | 72.1 $\downarrow_{6.1}$ | - |
| DP (Yang et al., 2023) | 94.9 $\downarrow_{0.7}$ | 93.8 $\downarrow_{1.8}$ | 90.8 $\downarrow_{4.8}$ | 77.2 $\downarrow_{1.0}$ | 73.1 $\downarrow_{5.1}$ | - |
| Random* | 94.8 $\downarrow_{0.8}$ | 94.5 $\downarrow_{1.1}$ | 93.0 $\downarrow_{2.6}$ | 77.3 $\downarrow_{0.9}$ | 75.3 $\downarrow_{2.9}$ | - |
| $\epsilon$-greedy Raju et al. (2021) | 95.2 $\downarrow_{0.4}$ | 94.9 $\downarrow_{0.7}$ | 94.1 $\downarrow_{1.5}$ | 76.4 $\downarrow_{1.8}$ | 74.8 $\downarrow_{3.4}$ | - |
| UCB Raju et al. (2021) | 95.3 $\downarrow_{0.3}$ | 94.7 $\downarrow_{0.9}$ | 93.9 $\downarrow_{1.7}$ | 77.3 $\downarrow_{0.9}$ | 75.3 $\downarrow_{2.9}$ | - |
| InfoBatch$\dagger$ Qin et al. (2024) | 95.6 $\uparrow_{0.0}$ | 95.1 $\downarrow_{0.5}$ | 94.7 $\downarrow_{0.9}$ | 78.2 $\uparrow_{0.0}$ | 78.1 $\downarrow_{0.1}$ | 76.5 $\downarrow_{1.7}$ |
| InfoBatch$\ddagger$ Qin et al. (2024) | 95.6 $\uparrow_{0.0}$ | 95.0 $\downarrow_{0.6}$ | 94.4 $\downarrow_{1.2}$ | 78.3 $\uparrow_{0.1}$ | 77.7 $\downarrow_{0.5}$ | $\backslash$ |
| RePB | 95.6 $\uparrow_{0.0}$ | 95.4 $\downarrow_{0.2}$ | 94.9 $\downarrow_{0.7}$ | 78.4 $\uparrow_{0.2}$ | 78.1 $\downarrow_{0.1}$ | 77.2 $\downarrow_{1.0}$ |

InfoBatch$\dagger$ 94.7%). This highlights RePB's ability to effectively prune data without significant performance degradation, showcasing the benefits of its consistency-preserving mechanisms.

**Large-Scale Vision-Language Tasks vs. InfoBatch.** We further benchmark RePB against InfoBatch on three challenging large-scale vision-language datasets: ToCa (zero-shot captioning), MJ+ST (scene text recognition), and SS1M (cross-domain captioning), as detailed in Table 2. Across these demanding tasks, RePB consistently demonstrates superior efficiency or performance. For instance, on ToCa (Table 2a), RePB achieves a higher pruning rate (35.8% vs. 34.1% for InfoBatch) while simultaneously outperforming both InfoBatch and full-dataset training on key metrics like NoCaps CIDEr (70.5 vs. 69.2 for InfoBatch, 70.5 for full). Similarly, for scene text recognition on the 15M-sample MJ+ST dataset (Table 2b), RePB prunes a substantial 44.4% of data—a greater reduction than InfoBatch (38.1%)—while maintaining performance equivalent to full training across all benchmarks (e.g., 96.1% on IIIT5k). On the SS1M cross-domain captioning task (Table 2c), RePB again achieves a higher pruning rate (34.8% vs. 23.0%) and yields improved or comparable CIDEr scores relative to both full training and InfoBatch. These results underscore RePB's enhanced effectiveness in handling large-scale, complex datasets where InfoBatch exhibits more modest pruning or comparatively lower performance. RePB's capacity for substantial data reduction while maintaining high fidelity highlights the robustness of its consistency-focused local pruning and temporal rescaling strategies.

Table 2: Comparative performance of RePB and InfoBatch on diverse large-scale vision-language and scene text recognition tasks. (a) Zero-shot captioning on 3M samples. (b) Scene text recognition on 15M samples. (c) Cross-domain captioning on 3M samples.

**(a) ViECap on ToCa Dataset**

| Method | Pruned % | NoCaps B@4 | NoCaps C | COCO B@4 | COCO C |
|---|---|---|---|---|---|
| Full | - | 26.6 | 70.5 | 27.1 | 95.2 |
| InfoBatch | 34.1 | 26.4 | 69.2 | 26.9 | 93.7 |
| RePB | 35.8 | 27.0 | 70.5 | 28.0 | 95.1 |

**(b) ABINet on MJ+ST**

| Method | Pruned % | IIIT5k | IC15 | SVTP | C80 |
|---|---|---|---|---|---|
| | | Accuracy % | | | |
| Full | - | 96.1 | 85.4 | 88.7 | 89.2 |
| InfoBatch | 38.1 | 95.9 | 84.2 | 88.0 | 88.4 |
| RePB | 44.4 | 96.1 | 85.3 | 89.1 | 89.2 |

**(c) ViECap on SS1M**

| Method | Pruned % | COCO B@4 | COCO C | Flickr30k B@4 | Flickr30k C |
|---|---|---|---|---|---|
| Full | - | 9.6 | 45.1 | 6.5 | 22.3 |
| InfoBatch | 23.0 | 9.2 | 44.4 | 6.4 | 22.4 |
| RePB | 34.8 | 9.5 | 46.3 | 6.7 | 22.6 |

## 5.3 Broad Generalization and Applicability of RePB

A key strength of a principled data pruning framework is its ability to generalize across diverse tasks, model architectures, and learning settings. We showcase RePB's wide-ranging applicability through an extensive set of experiments.

Table 3: Cross-architecture generalization on ImageNet-1K and CIFAR100 with CNNs, Transformers, and Mamba. RePB maintains accuracy while significantly reducing data across architectures.

| Method | ImageNet-1K | | | | | | CIFAR100 | |
| | CNN | | | Transformer | | Mamba | | |
| | R18 | R50 | EfficientNet | ViT | Swin | Vim | R18 | R50 |
|---|---|---|---|---|---|---|---|---|
| Full | 69.5 | 78.6 | 76.1 | 73.3 | 80.0 | 75.7 | 78.2 | 80.6 |
| +RePB | 69.5 / 32.1 | 78.5 / 30.2 | 76.1 / 28.0 | 73.3 / 23.3 | 80.0 / 38.3 | 75.6 / 31.3 | 78.3 / 39.6 | 80.8 / 49.5 |

**Cross-Architecture Generalization.** Table 3 demonstrates RePB's performance on ImageNet-1K and CIFAR100 across various architectural families: CNNs (ResNet18, ResNet50, EfficientNet), Transformers (ViT, Swin, ViECap), and Mamba-based models (Vim). RePB consistently achieves performance nearly identical to full dataset training while pruning a significant portion of data (e.g., 23-38% on ImageNet-1K, 38-50% on CIFAR100). This consistent, near-lossless performance across diverse architectures underscores RePB's model-agnostic nature, stemming from its fundamental approach to addressing consistency issues rather than relying on architecture-specific heuristics.

**Diverse Downstream Tasks.** Beyond standard classification, RePB exhibits strong performance across a variety of other vision and vision-language tasks (Tables 4, 5, 6). In multi-view stereo on WHU-MVS (Table 4), RePB with 37.3% pruning slightly improves MAE (0.1147m vs. 0.1185m) and coverage. Similarly, for cross-view geo-localization on CVACT (Table 5), RePB prunes 39.2% of data while maintaining or slightly improving recall metrics. For image captioning on COCO and video captioning on MSR-VTT (Table 6, bottom panel), RePB achieves improved or comparable results to full training with substantial pruning (e.g., 24.3% pruned on COCO captioning improving CIDEr; 48.1% pruned on MSR-VTT video captioning improving BLEU@4 and CIDEr). These results highlight RePB's versatility in adapting to tasks with complex objectives and data structures.

Table 4: Performance on the multi-view stereo task using the Ada-MVS model on the WHU-MVS dataset (28K image-depth maps).

| Method | Pruned % | MAE (m) $\downarrow\uparrow$ | <0.6m (%) $\uparrow$ |
|---|---|---|---|
| Ada-MVS | - | 0.1185 | 97.38 |
| RePB | 37.3 | 0.1147 | 97.54 |

Table 5: Performance on the cross-view geo-localization task using the GeoDTR model on the CVACT dataset (35K images).

| Method | Pruned % | R@1 | R@5 | R@10 | R@1% |
|---|---|---|---|---|---|
| GeoDTR | - | 86.21 | 95.44 | 96.72 | 98.77 |
| RePB | 39.2 | 86.21 | 95.68 | 96.90 | 98.74 |

Table 6: Performance of RePB on advanced vision-language tasks including image captioning and video captioning. RePB demonstrates strong performance with substantial data pruning.

**(a) Image Captioning on COCO**

| Method | Pruned % | NoCaps Val (CIDEr) | | | | COCO | | | |
| | | In | Near | Out | Overall | B@4 | M | C | S |
|---|---|---|---|---|---|---|---|---|---|
| ViECap | - | 58.4 | 63.1 | 65.3 | 65.2 | 27.1 | 24.6 | 91.5 | 18.0 |
| RePB | 24.3 | 58.4 | 63.7 | 65.9 | 65.8 | 27.6 | 24.6 | 92.8 | 18.0 |

**(b) Video Captioning on MSR-VTT**

| Pruned % | MSR-VTT | | | |
| | B@4 | M | C | S |
|---|---|---|---|---|
| - | 23.4 | 20.8 | 27.9 | 5.0 |
| 48.1 | 25.9 | 21.6 | 29.2 | 5.0 |

**Image Generation.** Table 7 shows RePB's application to image generation. For VAE on MNIST, DDPM on CIFAR10, and DDPM with Classifier Guidance on CIFAR10, RePB prunes 27-40% of the data while maintaining nearly identical FID scores compared to full dataset training. This demonstrates RePB's utility in generative modeling where data distribution fidelity is crucial.

Table 7: Performance in image generation tasks across different model architectures and datasets: VAE on MNIST, DDPM and DDPM with CFG on CIFAR10.

| Method | VAE | | DDPM | | DDPM-CFG | |
| | Pruned % | FID $\downarrow$ | Pruned % | FID $\downarrow$ | Pruned % | FID $\downarrow$ |
|---|---|---|---|---|---|---|
| Full | - | 35.34 | - | 16.38 | - | 14.89 |
| RePB | 39.8 | 35.33 | 27.3 | 16.22 | 27.3 | 14.80 |

**Supervised and Semi-Supervised Learning across Modalities.** Finally, Table 8 showcases RePB's effectiveness in standard supervised and semi-supervised learning scenarios across image, text, and

Table 8: Performance in (a) supervised and (b) semi-supervised learning across diverse modalities: image, text, and audio. RePB consistently maintains accuracy across paradigms and modalities.

**(a) Supervised learning**

| Method | $\text{Image}_{\text{EuroSAT}}$ ViT | | $\text{Text}_{\text{AG News}}$ BERT | | $\text{Audio}_{\text{ESC-50}}$ HuBERT | |
|---|---|---|---|---|---|---|
| | Pruned % | Acc | Pruned % | Acc | Pruned % | Acc |
| Full | - | 96.8 | - | 89.6 | - | 67.5 |
| RePB | 35.6 | 96.9 | 35.0 | 89.5 | 34.3 | 67.6 |

**(b) Semi-supervised learning**

| Method | $\text{Image}_{\text{CIFAR100}}$ FixMatch | | $\text{Text}_{\text{Yelp Review}}$ FlexMatch | | $\text{Audio}_{\text{ESC-50}}$ Dash | |
|---|---|---|---|---|---|---|
| | Pruned % | Acc | Pruned % | Acc | Pruned % | Acc |
| Full | - | 61.9 | - | 53.7 | - | 64.5 |
| RePB | 39.8 | 61.9 | 36.3 | 54.4 | 27.3 | 64.5 |

audio. In supervised learning (Table 8a), RePB applied to ViT (image), BERT (text), and HuBERT (audio) consistently prunes 35% of the data while maintaining or slightly improving accuracy. In semi-supervised learning (Table 8b), RePB integrated with FixMatch (image), FlexMatch (text), and Dash (audio) also demonstrates robust performance, achieving significant pruning (27-40%) with negligible or even positive impacts on accuracy. These results suggest that RePB's principles are broadly beneficial, irrespective of the learning paradigm or data modality.

In summary, the extensive experimental validation demonstrates that RePB not only outperforms existing dynamic data pruning methods on established benchmarks but also exhibits remarkable generalization capabilities across a wide spectrum of tasks, datasets, model architectures, and learning settings. Its ability to significantly reduce data requirements while preserving or even enhancing performance underscores its potential as a valuable tool for efficient and reliable machine learning.

**Compatibility with Sampling Strategies.** Beyond standard architecture and task generalization, we evaluate RePB's flexibility as a meta-framework by integrating it with recent importance sampling methods (Salaün et al., 2023; 2024). We conducted experiments on CIFAR-100 using ResNet18 under three configurations: (1) The original global optimal sampling strategy proposed in these works; (2) A hybrid approach where their resampling mechanism is applied locally within RePB's windows and reweighted by CTR; and (3) Standard RePB using gradient norm instead of loss. As shown in Table 9, integrating this external sampling strategy into the RePB framework yields consistent performance improvements

Table 9: Compatibility with alternative sampling and gradient norm on CIFAR-100 (ResNet18).

| Method | 30% | 50% | 70% |
|---|---|---|---|
| Salaun et al. | 77.7 | 77.3 | 76.7 |
| RePB + Salaun et al. | 78.2 | 77.7 | 77.2 |
| RePB (Grad Norm) | **78.6** | 78.0 | 77.0 |
| RePB (Loss) | 78.4 | **78.1** | **77.2** |

across all pruning rates (e.g., +0.5% accuracy at 30% pruning). This indicates that RePB's core mechanisms—mitigating score context drift via local windows and correcting temporal bias via CTR—provide structural benefits that complement even sophisticated sampling algorithms. Furthermore, RePB using gradient norm achieves performance comparable to the loss-based default. This demonstrates that RePB is metric-agnostic: it serves as a robust, general-purpose framework capable of enhancing various scoring criteria and selection strategies.

## 5.4 ABLATION STUDY

**Effect of each operation.** An ablation study on CIFAR100 (Table 10, R18/R50 at 30%/50% pruning rate) reveals the distinct contribution of each RePB component. Pruning without resampling (Res) results in sample pool depletion and performance collapse (19.0%/8.7%). Building on resampling, adding LWP to address score context drift boosts accuracy to 78.1%/80.2%, while adding CTR to mitigate temporal gradient bias achieves 78.2%/80.3%. The full RePB framework attains the highest performance, matching or exceeding full dataset training and substantially outperforming random pruning (Random*). This demonstrates that the principled strategies of LWP for valid score comparisons and CTR for temporal dynamic consistency are synergistically vital for achieving SOTA efficacy.

Table 10: Ablation of proposed operation on CIFAR100.

| Operation | | | Accuracy | |
|---|---|---|---|---|
| *Res* | *LWP* | *CTR* | R18 | R50 |
| | ✓ | ✓ | 19.0 | 8.7 |
| ✓ | | | 77.5 | 79.9 |
| ✓ | ✓ | | 78.1 | 80.2 |
| ✓ | | ✓ | 78.2 | 80.3 |
| ✓ | ✓ | ✓ | 78.4 | 80.8 |
| Random* | | | 77.3 | 79.7 |
| Full Dataset | | | 78.2 | 80.6 |

**Impact of Window Size.** Figure 2 compares RePB (varying window sizes $W$, multiples of $B = 128$) against InfoBatch (Annealing) and InfoBatch* (No Annealing, for higher pruning limits) on CIFAR100 (ResNet50). RePB consistently outperforms both InfoBatch variants across most pruning percentages, regardless of its window size. For instance, at 40% pruning, all RePB configurations achieve $\sim$78%+ accuracy, surpassing InfoBatch ( 77.8%) and InfoBatch* ( 78.0%), with this advantage widening at higher pruning rates (50-70%). Within RePB, smaller windows ($W = 1B$ to $4B$) generally yield optimal performance, particularly up to 55% pruning. The $W = 1B$ setting, which theoretically eliminates score context drift by using scores from a single pre-update batch, often defines the upper performance envelope, validating our core hypothesis. As $W$ increases (e.g., $W \geq 16B$), a marginal performance decrease is observed, suggesting that even slight

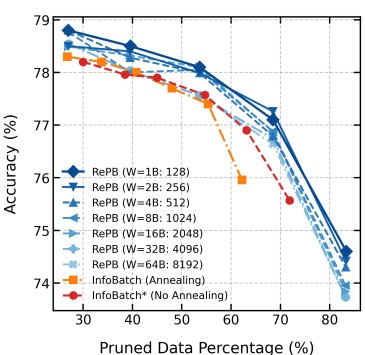

Figure 2: Impact of window size.

increases in potential intra-window drift can have a subtle impact, further underscoring the benefit of RePB's local context approach. Crucially, even RePB with larger windows largely maintains superiority over the global comparison methods of InfoBatch. This indicates RePB's relative robustness to the window size hyperparameter, and consistently affirms the advantage of its principled local pruning strategy over global score aggregation.

## 6  CONCLUSION

In this work, we addressed two critical consistency challenges in dynamic data pruning: *score context drift*, which invalidates inter-batch importance comparisons, and *temporal gradient bias*, which skews training dynamics. We proposed RePB, a framework that systematically resolves these issues by integrating *local window pruning* for valid score comparisons within stable model contexts, and *cumulative temporal rescaling* to align the expected gradient trajectory with full-dataset training via inverse historical sampling frequency weighting. Extensive empirical validation across diverse datasets, tasks (classification, vision-language, generation), and architectures (CNNs, Transformers, SSMs) demonstrates RePB's consistent superiority over state-of-the-art methods like InfoBatch, often matching or exceeding full-dataset performance with significant data reduction. By offering a theoretically grounded solution to previously underappreciated consistency violations, RePB establishes a more principled, robust, and effective paradigm for dynamic data pruning, enabling more reliable and efficient training of various models across diverse applications.

ACKNOWLEDGMENTS

This work was supported by the National Natural Science Foundation of China under Grant 62471394, U21B2041, 62306241 and 62576284.

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

# A    DETAILED COMPARISON WITH INFOBATCH RESCALING

While both RePB and InfoBatch (Qin et al., 2024) employ a form of loss/gradient rescaling in conjunction with dynamic data pruning, their rescaling mechanisms, theoretical motivations, and practical implications differ significantly. This appendix elucidates these distinctions, focusing on RePB's Cumulative Temporal Rescaling (CTR) versus InfoBatch's instantaneous expectation rescaling.

## A.1    INFOBATCH: INSTANTANEOUS UNBIASED GRADIENT ESTIMATION

InfoBatch aims to achieve an *unbiased estimate of the full batch gradient at each training step* $t$. As described in their work (see Figure 1 and Section 2.3 of Qin et al. (2024)), given a dataset (or a current epoch's data pool), InfoBatch maintains scores (e.g., loss values) for samples. It then soft prunes by stochastically discarding a portion of low-score samples. Specifically, if a sample $z$ has a score $H_t(z)$ below a threshold $\mathcal{H}_t$ (e.g., mean loss) and its pruning probability is $r \in (0, 1)$, its gradient is scaled by $1/(1 - r)$ if it is kept. Samples with scores $H_t(z) \geq \mathcal{H}_t$ are not modified (implicitly, their pruning probability is 0, so $1/(1 - 0) = 1$).

The core objective of InfoBatch's rescaling is to ensure that the *expected gradient* computed on the pruned subset $S_t$ at step $t$ approximates the gradient that would have been computed on the original (pre-pruning) data pool $D_t$ for that step:

$$\mathbb{E}_{z \sim S_t, P_t(z)} \left[ \gamma_t(z) \nabla \ell(z, \theta_t) \right] \approx \mathbb{E}_{z \sim D_t} \left[ \nabla \ell(z, \theta_t) \right] \tag{10}$$

where $P_t(z)$ is the probability of pruning sample $z$ at step $t$ (if $H_t(z) < \mathcal{H}_t$, $P_t(z) = r$; otherwise $P_t(z) = 0$), and $\gamma_t(z) = 1/(1 - P_t(z))$ is the rescaling factor. The expectation is taken over the stochastic pruning decisions at step $t$, conditioned on the model $\theta_t$ and the scores $\{H_t(z)\}$. This strategy focuses on making each individual gradient update step an unbiased estimator relative to the data available for pruning *at that instant*. The pruning probabilities and rescaling factors are typically determined based on the current batch or epoch's score distribution and are not explicitly dependent on the long-term selection history of individual samples across multiple epochs. InfoBatch also employs an annealing schedule, training on the full dataset in the last few epochs to reduce variance and potential remaining bias.

## A.2    REPB: CUMULATIVE TEMPORAL RESCALING FOR LONG-TERM DYNAMIC STABILITY

RePB's Cumulative Temporal Rescaling (CTR) mechanism serves a different primary purpose: to ensure the *long-term expected cumulative gradient dynamics* of training with pruned data align with those of training on the full dataset uniformly over the entire training trajectory. It addresses the *temporal bias* introduced by the fact that dynamic pruning causes different samples to be seen with varying frequencies *across epochs*.

In RePB, each sample $(x_i, y_i)$ in the selected batch $S'_t$ at a global iteration $k$ (where $k$ typically corresponds to an epoch or a longer period over which rescaling weights are updated) has its loss $\ell(x_i, y_i, \theta_t)$ scaled by $w_{i,k} = 1/f_{i,k}$, where $f_{i,k} = c_i/k$ is the empirical cumulative selection frequency of sample $i$ up to iteration $k$ (i.e., selected $c_i$ times in $k$ rescaling periods). The rescaled loss is $L_{RePB}(S'_t, \theta_t) = \frac{1}{|S'_t|} \sum_{i \in S'_t} w_{i,k} \cdot \ell(x_i, y_i, \theta_t)$.

The key differences and advantages of RePB's CTR are:

1. **Addressing Long-Term Temporal Bias:** CTR explicitly accounts for the entire selection history of each sample. If a sample has been historically under-selected relative to a uniform draw from the dataset, its contribution is up-weighted when it finally appears in a batch. This aims to correct for biases not just within a single pruning step or epoch, but over the entire course of training. InfoBatch's rescaling, being instantaneous, does not inherently correct for such long-term under- or over-representation of specific samples across epochs.

2. **Stabilizing Cumulative Gradient Magnitude:**  By  using  $w_{i,k} \approx$ $1/($avg. probability of seeing sample $i)$, CTR aims to make the *expected total gradient signal* contributed by each sample over many epochs proportional to what it would have contributed if seen uniformly. This helps to maintain consistent training dynamics

and prevents the learning trajectory from being unduly skewed by the pruning process itself. The goal is $E[\sum_t \Delta \theta_t^{RePB}] \approx E[\sum_t \Delta \theta_t^{Full}]$, ensuring the overall learning path remains faithful to full-dataset training in terms of expected update magnitudes.

3. **Independence from Instantaneous Pruning Probabilities:** RePB's CTR relies on observed historical frequencies rather than needing to precisely define or estimate the probability $P_t(z)$ with which a sample is pruned at the current step $t$ for the purpose of calculating an inverse probability weight. This can be an advantage because in complex dynamic pruning schemes (especially with local windowing like RePB's LWP), the exact instantaneous selection probability of a given sample can be difficult to model precisely, as it depends on the composition of its current local window and the model state. CTR's empirical, historical approach sidesteps this modeling challenge.

4. **Synergy with Local Window Pruning (LWP):** RePB's LWP component addresses score context drift by ensuring score comparisons are valid within a local window. CTR then complements this by addressing the longer-term consequences of these local (but still potentially non-uniform over the whole dataset) selections. InfoBatch, typically performing epoch-wide score comparison for pruning, faces the score context drift issue that LWP is designed to solve. While its rescaling aims for unbiasedness relative to that epoch's (potentially flawed) pruning decision, it doesn't correct the upstream issue of score incomparability within the epoch, nor the downstream issue of long-term sample frequency bias in the same direct way as CTR.

## A.3 Illustrative Example

Consider a scenario where, due to the dynamics of the model and data, a specific subset of informative samples (e.g., hard negatives) is consistently assigned low loss scores for several epochs by InfoBatch's global scoring, leading to them being frequently pruned (even if stochastically). While InfoBatch's rescaling would make the gradient unbiased for the steps where these samples are selected, it doesn't inherently compensate for their overall reduced exposure across these epochs. In contrast, RePB's CTR would track that these samples have low $c_i$ values. When they are eventually selected (perhaps due to resampling or changes in their loss within a local window), their $w_{i,k}$ would be high, significantly boosting their contribution to compensate for past under-selection. This helps ensure their cumulative impact on the model is not diminished over the long term. Furthermore, LWP makes it more likely that their true importance (relative to their local peers) is accurately assessed when they do appear in a window, avoiding premature dismissal based on comparison with globally easier samples.

## A.4 Summary of Advantages of RePB's Approach

In summary, RePB's dual strategy offers distinct advantages:

- **LWP ensures valid score comparisons**, mitigating score context drift, a problem not directly addressed by InfoBatch's epoch-wide score aggregation for pruning.

- **CTR provides long-term dynamic stability**, correcting for temporal biases in sample exposure across epochs, which is a more holistic approach than InfoBatch's focus on instantaneous unbiasedness at each pruning step.

- **CTR is empirically driven by historical data**, making it robust and not reliant on explicit modeling of complex, instantaneous selection probabilities.

These aspects contribute to RePB's strong performance and generalization, particularly its ability to maintain high accuracy at significant pruning rates across diverse and large-scale tasks, as demonstrated in our experiments. While InfoBatch's annealing (training on full data at the end) can recover some performance lost due to earlier biases, RePB aims to maintain a more consistent and less biased training trajectory throughout the pruning phase itself.

## B  Efficiency Metric

### B.1  Computational Efficiency

Reproducing exact wall-clock time reductions for dynamic data pruning methods can be notoriously challenging due to variations in hardware, software implementations, and system-level optimizations. We therefore primarily adopt the **percentage of data pruned (Pruned %)** as our main efficiency metric, reflecting the reduction in samples processed. This choice is strongly supported by the minimal computational overhead introduced by RePB.

As shown in Table 11, which details the overhead for processing 1 million samples on an NVIDIA RTX 3090 GPU, RePB's pruning logic is exceptionally lightweight. Its overhead is measured at a mere 0.082 seconds, which is even lower than that of InfoBatch (0.236s). More critically, this overhead is negligible when compared to the backbone processing time for standard architectures. For ResNet18 (734.1s for 1M samples), RePB's overhead constitutes only 0.011% of the backbone time, and for ResNet50 (2122.4s), this ratio drops further to an almost imperceptible 0.004%. This is substantially lower than InfoBatch's ratios (0.032% for ResNet18 and 0.011% for ResNet50).

This extremely low overhead ensures that RePB is compute-positive (Evans et al., 2024); the computational cost of executing RePB's pruning and rescaling logic is vastly outweighed by the significant savings achieved from processing a reduced dataset (typically ¿20-30% pruned in our experiments, often much higher). For more complex models or those with larger, slower backbones than ResNet18/50, or tasks involving multiple intricate components beyond a single backbone (e.g., in large vision-language models), the relative pruning overhead of RePB would be even smaller, further

Table 11: Computational overhead of RePB (1M samples, NVIDIA RTX 3090 GPU). Overhead is the time for the pruning logic itself. Backbone Time is provided for ResNet18 (R18) and ResNet50 (R50) for processing 1M samples. P/B is the ratio of Overhead to Backbone Time.

| Method | Overhead | R18 P/B 734.1s | R50 P/B 2122.4s |
|--------|----------|----------------|-----------------|
| InfoBatch | 0.236s | 0.032% | 0.011% |
| RePB | 0.082s | 0.011% | 0.004% |

amplifying its efficiency benefits. Consequently, the Pruned % serves as a robust, hardware-agnostic, and readily comparable indicator of the substantial computational savings and training acceleration potential offered by RePB.

## C  Limitations and Future Work

While RePB robustly addresses key consistency challenges in dynamic pruning, future work could explore several refinements. The optimal local window size ($W$) for LWP, though RePB shows good tolerance, might benefit from adaptive selection based on dataset or training phase characteristics. Similarly, while CTR effectively provides long-term gradient alignment, its adaptation speed in highly non-stationary scenarios could be investigated, potentially by incorporating short-term bias indicators. Finally, extending LWP to incorporate diverse or multiple importance metrics beyond sample loss presents an interesting avenue for enhancing its decision-making process. These directions aim to further broaden RePB's applicability and refine its efficiency while building upon its demonstrated strengths in principled, consistent data pruning.

## D  The Use of Large Language Models (LLMs)

The authors affirm that this manuscript was written primarily by human authors. However, sections of this paper underwent language refinement and grammatical corrections with the assistance of a LLM, specifically Gemini-2.5 Pro. This tool was used exclusively for improving clarity and readability and did not contribute to the scientific content, methodology, or core conclusions of the research. All scholarly responsibility for the content remains with the human authors.

