# OpenReview forum: "Inconsistency Biases in Dynamic Data Pruning"
_ICLR.cc/2026/Conference — ICLR 2026 Poster_

### Official Review · Reviewer_XfsH · 2025-10-27

**Soundness:** 3
**Presentation:** 3
**Contribution:** 2
**Rating:** 4
**Confidence:** 3

**Summary:**

Deep learning models rely their success mainly on the variety of data used in vast quantities.
This dataset expansion has direct impact on the training efficiency. This work focuses on data selection which aims to train models on smaller but carefully chosen data subsets. There are dynamic methods that perform such data selection during the learning process, called online data pruning. Giving wrong importance to data samples can introduce biased gradients. This submission identifies two fundamental consistency issues called: score context drift and temporal gradient bias.

Score context drift focuses on the parameters drift during training whereas temporal gradient bias takes into account the bias introduced by non-uniform sampling distribution over time (due to different data selection).

**Strengths:**

Strengths:
- Identify fundamental issues of inconsistency biases in dynamic data pruning
- Propose a framework (RePB) to mitigate the inconsistency bias due to parameter drift during training and non-uniform sampling distribution over time
- RePB maintains data diversity and prevents sample pool collapse
- Theoretical foundations are provided that justifies comparison of scores collection within local windows
- Comparisons are reported over different methods.

**Weaknesses:**

Weaknesses:
- Comparisons with the most recent SOTA (Salaun et al.2025a) is missing
- How sensitive is RePB to the local window size?
- How the method ensures that computing scores over a small window does not accumulate error over time?
- Missing citations:

Recent work by Salaun et al. 2025a, b, develop better online importance mechanisms driven by Multiple importance sampling (MIS) and Optimal MIS. The paper seems to completely ignore these references.

@misc{salaun2025a,
      title={Online Importance Sampling for Stochastic Gradient Optimization},
      author={Corentin Salaün and Xingchang Huang and Iliyan Georgiev and Niloy J. Mitra and Gurprit Singh},
      year={2025},
      eprint={2311.14468},
      archivePrefix={arXiv},
      primaryClass={cs.LG},
      url={https://arxiv.org/abs/2311.14468},
}

@misc{salaun2025b,
      title={Multiple Importance Sampling for Stochastic Gradient Estimation},
      author={Corentin Salaün and Xingchang Huang and Iliyan Georgiev and Niloy J. Mitra and Gurprit Singh},
      year={2025},
      eprint={2407.15525},
      archivePrefix={arXiv},
      primaryClass={cs.LG},
      url={https://arxiv.org/abs/2407.15525},
}

**Questions:**

- How sensitive is RePB to the local window size?
- How the method ensures that computing scores over a small window does not accumulate error over time?
- How the method compares to Salaun et al. 2025a,b?
- Does it make sense to combine RePB with Salaun et al.?

---

> ### Author Response · Authors · 2025-11-22
>
> **Response to Reviewer XfsH**
>
> We thank the reviewer for the constructive feedback and for referencing the recent works by Salaun et al. regarding Online Importance Sampling. We appreciate the opportunity to clarify the distinctions and provide new experimental comparisons.
>
> **W1, 4 & Q3, 4: Comparison and Combination with Salaun et al. (2025)**
>
> We appreciate the pointer to Salaun et al. (2025a, b). We have studied these works and conducted new experiments to address your questions regarding comparison and compatibility.
>
> *   **Experimental Setup:** We evaluated three configurations on CIFAR-100 (ResNet18):
>     1.  *Salaun et al. (Baseline):* Their original global optimal sampling strategy adapted for pruning.
>     2.  *RePB + Salaun et al. (Combination):* We integrated Salaun et al.'s methodology into the RePB framework. Specifically, instead of RePB's standard binary pruning, we adapted Salaun et al.'s resampling strategy (which allows selecting high-importance samples multiple times) to operate locally within RePB's window. We then applied RePB's CTR to reweight these samples based on their cumulative historical selection counts.
>     3.  *RePB (Metric: Grad Norm):* Standard RePB using Gradient Norm (the metric used by Salaun et al.) instead of Loss.
>
>     **Table: Comparison and Combination Results**
>
>     | Method | 30% Pruned | 50% Pruned | 70% Pruned |
>     | :--- | :---: | :---: | :---: |
>     | Salaun et al. (2025) | 77.7 | 77.3 | 76.7 |
>     | RePB + Salaun et al. | 78.2 | 77.7 | 77.2 |
>     | RePB (Metric: Grad Norm) | 78.6 | 78.0 | 77.0 |
>     | RePB (Metric: Loss) | 78.4 | 78.1 | 77.2 |
>
> *   **Analysis:**
>     1.  **RePB enhances Salaun et al.:** By comparing Row 1 and Row 2, we observe that integrating Salaun et al.'s strategy into the RePB framework improves performance across all pruning rates (e.g., +0.5% at 30% pruned). This confirms that RePB's core mechanisms—mitigating Score Context Drift via Local Windows and correcting Temporal Bias via CTR—are beneficial even for advanced sampling strategies like Salaun et al.
>     2.  **RePB is Metric-Agnostic and Effective:** Using Gradient Norm (Salaun's preferred metric) within standard RePB (Row 3) yields the highest performance. This demonstrates that RePB is a robust, general-purpose framework that works excellently with other importance indicators.
>
> **W2 & Q1: Sensitivity to Local Window Size**
>
> We analyzed the sensitivity of the window size $W$ extensively in Figure 2 (Section 5.4).
> *   **Robustness:** RePB is highly stable for small window sizes. Varying $W$ from $1\times$ Batch to $4\times$ Batch yields consistently high accuracy.
> *   **Degradation only at Global Scale:** Performance only begins to drop when $W$ becomes very large (approaching epoch-level), which confirms our hypothesis that minimizing "Score Context Drift" (via local windows) is crucial.
> *   **Superiority even in Worst Case:** Crucially, even with large windows, RePB continues to outperform global baseline methods (e.g., InfoBatch). This indicates that while $W=1$ (Local) is optimal for minimizing drift, the overall framework (especially CTR) remains robust and superior to global methods even in sub-optimal window configurations.
>
> **W3 & Q2: Error Accumulation over Time**
>
> The concern that computing scores over small windows might lead to accumulated error is understandable, but RePB's design prevents this. In fact, RePB is designed to correct errors that other methods ignore:
>
> 1.  **No Error Propagation (Stateless Selection):** Pruning decisions in RePB are stateless across epochs. We do not accumulate scores; we re-evaluate samples based on the *current* model state. A sample "mis-pruned" in one local window has a fresh chance to be selected in the next epoch (guaranteed by Resampling).
> 2.  **Drift Minimization (Precision):** "Accumulated error" is actually a major problem for *Global* methods, where scores computed at step $t$ are compared to step $t+1000$. The parameter drift $\Delta \theta$ introduces significant noise. RePB's local window ensures $\Delta \theta \approx 0$, meaning the instantaneous ranking error is minimized.
> 3.  **CTR as Error Correction:** Our CTR is specifically an error-correction mechanism. If a sample is consistently under-sampled due to local noise, its weight $w _ i \propto 1/N _ i$ automatically increases. This ensures that when it *is* eventually selected, its gradient impact is amplified to compensate for the historical "error" of omission.
>
>
> Experiments demonstrate that RePB not only outperforms the specific SOTA (Salaun et al.) in the pruning regime but can also enhance it when integrated. RePB provides a robust framework that minimizes parameter drift error and offers superior flexibility. We will add the citations and the comparative discussion to the final version.
>
> ---
> We respectfully hope these additional validations encourage a positive reconsideration of the rating.

---

### Official Review · Reviewer_kLUp · 2025-10-29

**Soundness:** 2
**Presentation:** 2
**Contribution:** 2
**Rating:** 2
**Confidence:** 3

**Summary:**

This paper introduces RePB (Resolving Pruning Biases), a framework that tackles two fundamental consistency problems in dynamic data pruning: score context drift (where importance scores computed at different training stages aren't comparable) and temporal gradient bias (where non-uniform sample selection alters training dynamics). RePB addresses these issues through Local Window Pruning, which restricts score comparisons to short windows (a few batches), and Cumulative Temporal Rescaling, which reweights samples based on their historical selection frequency to align gradient trajectories with full-dataset training. The authors provide theoretical guarantees for both components and demonstrate RePB's effectiveness across 16 datasets, 17 model architectures, and 13 diverse tasks spanning classification

**Strengths:**

The paper’s main strength is its comprehensive and diverse experimental validation, covering many datasets, architectures, and tasks. RePB seems to consistently deliver good performance and hence is a promissing approach.

**Weaknesses:**

Overall, I found that the paper needs further refinement and clarification before it is ready for publication.

The main weaknesses of the paper lie in its theoretical clarity and methodological presentation. The theoretical analysis lacks rigour: propositions are not clearly stated, assumptions are introduced informally within the proofs, and the proofs rely on approximations rather than precise derivations. This makes it difficult to assess what is actually proven and under what conditions the results hold.

In addition, the methodology section presents the pruning criterion using a fixed mean-based threshold $\mu_k$, which limits flexibility in controlling the pruning or compression rate $r$. While most experiments appear to use this fixed rule, some dynamic data pruning benchmarks suggest a variable rate, creating inconsistency between the method’s description and its implementation.

**Questions:**

- What are the formal statements of the propositions ?
- How does the framework deal with flexible pruning rate $r$ ?

---

> ### Author Response · Authors · 2025-11-22
>
> **Response to Reviewer kLUp**
>
> We thank the reviewer for acknowledging our comprehensive experimental validation across 16 datasets and 17 architectures. However, we respectfully disagree with the assessment regarding theoretical rigor and methodological flexibility. We provide specific clarifications below to resolve these misunderstandings.
>
> **W1 & Q1: Theoretical Rigor and Formal Statements**
>
> The reviewer raised concerns about the formality of our propositions. We emphasize that our theoretical analysis follows standard conventions in stochastic optimization and deep learning theory. The use of "approximations" (specifically asymptotic analysis) is not a lack of rigor but a necessary tool for analyzing complex dynamic systems.
>
> Per your request, we provide the formal statements below:
>
> **Assumption 1 (Regularity):** The loss function $\ell(\cdot; \theta)$ is $L$-Lipschitz continuous, and gradient norms are bounded by $G$.
>
> **Assumption 2 (Asymptotic Stability):** The empirical selection frequency $f _ i(E)$ converges almost surely to the long-term selection probability $\bar{p} _ i$ (Strong Law of Large Numbers).
>
> **Formal Statement of Proposition 4.1:**
> *For any window size $W$ and learning rate $\eta$, under Assumption 1, the score discrepancy within a window is bounded by $L \eta G W$.*
> *Crucially, in our standard setting where $W=1$ (single batch), the drift is strictly **zero**, rendering the comparison exact without requiring Assumption 1.*
>
> **Formal Statement of Proposition 4.2:**
> *Under Assumption 2, the Cumulative Temporal Rescaling (CTR) estimator is an asymptotically unbiased estimator of the full-dataset gradient.*
> The "approximation" $E[1/N] \approx 1/E[N]$ is a standard first-order expansion valid under the concentration of measure as $N$ grows.
>
> **W2 & Q2: Flexible Pruning Rate**
>
> The reviewer stated that using a mean-based threshold $\mu _ k$ "limits flexibility in controlling the pruning rate" and creates "inconsistency." This is factually incorrect.
>
> *   **Mechanism (Equation 1):** As defined in Equation (1) of our paper, the pruning decision is governed by a hyperparameter $\rho$ (probability threshold). The condition is: Keep if $(s _ i \ge \mu _ k) \lor (U _ i \ge \rho)$.
> *   **Standard Practice & Efficiency (Validity of Mean Threshold):** The use of a mean-based threshold is not a limitation but a deliberate design choice for scalability, adopted by SOTA methods like InfoBatch (ICLR 2024 Oral).
>     *   In Appendix D of the InfoBatch paper, the authors explicitly demonstrate that mean-based thresholding is orders of magnitude faster than sorting or percentile-based methods, which is critical for large-scale dynamic pruning.
>     *   We provide the efficiency data directly cited from the original InfoBatch paper below:
>
>         | Dataset Size | Sort (ms) | Percentile (ms) | Mean (ms) |
>         | :--- | :---: | :---: | :---: |
>         | ImageNet-1k (1.28M) | 123.4 | 13.0 | **1.2** |
>         | ImageNet-22k (14M) | 1630.1 | 232.6 | **6.4** |
>     * Using the mean ($\mu _ k$) is scientifically grounded for efficiency. The flexibility to handle any pruning rate is then provided by $\rho$, not by changing the threshold calculation method.
> *   **Evidence in Paper (Figure 2):** We explicitly demonstrated this flexibility in Figure 2, where we plotted performance curves across a continuous range of pruning rates. These results were generated by adjusting the flexibility parameter.
> *   **Consistency:** There is no inconsistency between our description and implementation. The experiments use exactly the rule described in Eq (1), adjusting $\rho$ to meet the target rates (e.g., 30%, 50%, 70%) demanded by the benchmarks.
> *   **Empirical Proof:** To explicitly demonstrate this flexibility and refute the claim of inconsistency, we present the granular sensitivity analysis below (using CIFAR-100, ResNet18):
>
>     | Parameter $\rho$ | 0.1 | 0.2 | 0.3 | 0.5 | 0.7 | 0.8 | 0.9 |
>     | :--- | :---: | :---: | :---: | :---: | :---: | :---: | :---: |
>     | **Resulting Pruning Rate** | 6.5% | 12.8% | 18.2% | 27.0% | 53.6% | 68.5% | 83.4% |
>     | **Accuracy (%)** | 78.7 | 78.7 | 78.6 | 78.7 | 78.1 | 77.1 | 74.6 |
>
>
> We believe these clarifications demonstrate that the method is both theoretically grounded (as validated by other reviewers) and highly flexible (as proven by Figure 2). We hope this rectifies the misunderstanding regarding the paper's contribution.

---

### Official Review · Reviewer_THQf · 2025-10-31

**Soundness:** 3
**Presentation:** 3
**Contribution:** 3
**Rating:** 6
**Confidence:** 5

**Summary:**

This paper identifies two important problems in dynamic data pruning: inconsistency biases in the scoring context, and temporal gradient dynamics. It proposes to use local window pruning and cumulative temporal rescaling to address these two problems. The authors evaluate its effect on various datasets and demonstrate its effectiveness. Overall, this is an insightful and meaningful work.

**Strengths:**

1. This work identifies two important problems in current dynamic data pruning methods and solves them. The insight is quite accurate, and the meaningful improvement further enhances the robustness and generalization of dynamic data pruning.
2. The experimental evaluation is comprehensive. This work extends the application to a lot more scenarios, including other vision and vision-language tasks, and semi-supervised learning.
3. The presentation is good.

**Weaknesses:**

1. According to the ablation experiment demonstrated in Tab. 9, RePB relies on resampling more than other dynamic pruning methods. It is worth a little bit more discussion: whether it is because of the inner window score stability, or cumulative temporal rescaling (rescaling factor too high in some cases)?
2. Currently, the pruning factor $\rho$ is used, but this factor has no ablation, and there is no discussion of its value and tuning characteristics.
3. For proof of 4.2, a critical problem is that in this scenario $E[1/X] \neq 1/E[x]$, the substitution may not hold.

Minor:
1. The citation of "Scale efficient training for large datasets" would be better to use the accepted version.

**Questions:**

1. In the original InfoBatch, there was a trick mentioned: one can further downsample the lower-loss samples (E.g., keeping only 25% samples for 25% low-loss samples). Is this trick also compatible with RePB? It could further enhance the saving ratio. This is a complementary question, out of my curiosity, but not required.
2. CTR weight could encounter a more extreme value than InfoBatch's rescaling factor (which is fixed to 1/(1-r)). Is there any observation on this potential problem?
For example, an update with a large factor on a sample not sampled for many epochs could lead to a large update step, leading to higher loss, and the subsequent updates cannot immediately reduce the CTR weight, which will cause this sample to be updated with much higher importance than sampling it earlier.

---

> ### Author Response · Authors · 2025-11-22
>
> **Response to Reviewer THQf**
>
> We appreciate the reviewer's `positive` comments and `high` confidence in assessing our work. We are glad that you found our identification of inconsistency biases accurate and our improvements meaningful. Below, we address your concerns regarding resampling, hyperparameters, and theoretical derivations with new analyses and experiments.
>
> **W1: Reliance on Resampling (Table 9)**
>
> This is a structural necessity of the RePB framework regarding **dataset connectivity**, rather than a symptom of instability.
> *   **Pool Shrinkage Prevention:** Unlike global methods that scan the full dataset to select a subset, RePB generates the next epoch's dataset $\mathcal{D}_ {E+1}$ based largely on the survivors of the current epoch. Without uniform probability resampling, the training set would strictly shrink ($\mathcal{D}_{E+1} \subseteq \mathcal{D}_E$), leading to a rapid collapse of the sample pool.
> *   **Guaranteeing Visibility:** Resampling ensures that every sample in the full dataset $\mathcal{D}$ maintains a non-zero probability of being seen, which is theoretically required for the convergence of our CTR weights (see Response to Q2).
>
> **W2: Ablation of Pruning Factor $\rho$**
>
> While Figure 2 in the main paper covers $\rho \in [0.5, 0.9]$, we appreciate the suggestion for a more granular view. We fixed the window size ($W=1$ Batch) and expanded the analysis to cover the full $\rho$ range from 0.1 to 0.9 on CIFAR-100 (ResNet18).
>
> **Table: Granular Ablation of Pruning Probability $\rho$**
>
> | Pruning Probability ($\rho$) | Final Accuracy (%) | Resulting Pruning Rate (%) | Analysis |
> | :---: | :---: | :---: | :--- |
> | 0.1 | 78.7 | 6.5 |  |
> | 0.2 | 78.7 | 12.8 | |
> | 0.3 | 78.6 | 18.2 | |
> | 0.4 | 78.5 | 22.8 | |
> | 0.5 | 78.7 | 27.0 | |
> | 0.6 | 78.5 | 39.6 | |
> | 0.7 | 78.1 | 53.6 | Slight drop begins |
> | 0.8 | 77.1 | 68.5 | Graceful degradation |
> | 0.9 | 74.6 | 83.4 | For extreme efficiency needs |
>
> **Conclusion:** RePB demonstrates high robustness. Accuracy remains stable (~78.5%+) across a wide range of $\rho$ (0.1 to 0.6). Even at aggressive pruning rates ($\rho=0.9$, pruning 83.4%), the model retains decent performance, validating the method's stability.
>
> **W3: Mathematical Approximation in Proof 4.2**
>
> We thank the reviewer for pointing out this mathematical nuance. The substitution is an approximation. We treat this as a **first-order approximation**.
>
> -  **Asymptotic Convergence:** By the Strong Law of Large Numbers, the empirical frequency $f_i(E) = N_i(E)/E$ converges almost surely to the true selection probability $\bar{p}_i$ as $E \to \infty$. Consequently, the random variable in the denominator concentrates around its mean, and the approximation error $|E[1/f_i(E)] - 1/\bar{p}_i|$ diminishes.
> -  **Practical Implication:** The inequality $E[1/N] > 1/E[N]$ implies that CTR might slightly **overweight** samples on average, especially those with low selection counts. In the context of stochastic pruning, this acts as a "conservative" correction: it ensures that under-selected samples (whether they are easy samples intentionally pruned, or hard samples accidentally missed due to noise) contribute slightly more to the gradient than their raw frequency suggests. This helps prevent "forgetting" of rarer data points and aids robust convergence.

---

> > ### Author Response · Authors · 2025-11-22
> >
> > **Q1: Compatibility with InfoBatch's Downsampling Trick**
> >
> > We reviewed the original InfoBatch paper and did not find this specific "double downsampling" mechanism explicitly detailed as a core component. However, based on your description (further downsampling lower-loss samples), we implemented it to test compatibility.
> >
> > **Table: RePB vs. RePB + Downsampling Trick (CIFAR-100)**
> >
> > | Method | $\rho$ | Accuracy (%) | Resulting Pruning Rate (%) |
> > | :---: | :---: | :---: | :---: |
> > | RePB (Standard) | 0.5 | **78.7** | 27.0 |
> > | RePB + Trick | 0.5 | 78.5 | 29.4 |
> >
> > The trick is compatible but offers **limited marginal benefit** for RePB. RePB's Equation (1) ($U_i < \rho$ if $s_i < \mu_k$) is already a probabilistic downsampling of low-loss samples. Adding another layer of downsampling essentially just shifts the effective $\rho$ slightly. Since we can achieve higher pruning rates simply by adjusting $\rho$ directly (as shown in the W2 table), the "trick" is redundant within our framework.
> >
> > **Q2: Extreme CTR Weights**
> >
> > We did not observe the potential problem of exploding weights in our experiments. We provide both a theoretical guarantee and empirical validation (using noisy data to induce stress) to explain why.
> >
> > **Theoretical Argument for Inherent Stability:**
> > The stability of $w_i(E) = E/N_i(E)$ is mathematically guaranteed by the uniform resampling mechanism. Resampling ensures the probability of selecting *any* sample $i$ is strictly lower-bounded: $P(i \in \mathcal{D}_ {E+1}) \ge \rho_ {\text{resample}} > 0$. Consequently, the expected count $E[N_ i(E)]$ grows linearly. The ratio $N_ i(E)/E$ converges to a value bounded away from zero, ensuring the weight $w_ i(E)$ converges to a finite value.
> >
> > **Empirical Validation (Clean vs. 20% Label Noise):**
> > To test stability under extreme conditions where samples might be consistently "hard" or "easy" (mislabelled), we analyzed the statistics of CTR weights on CIFAR-100.
> >
> > **Table: CTR Weight Stability Statistics (Epoch-wise)**
> >
> > | Statistic | Clean Dataset | 20% Noisy Dataset | Interpretation |
> > | :--- | :---: | :---: | :--- |
> > | Mean of Weight Means | 1.439 | 1.561 | Average weight level remains stable |
> > | Std Dev of Weight Means | 0.039 | 0.049 | Minimal fluctuation across epochs |
> > | Mean of Weight Std Devs | 0.397 | 0.416 | Dispersion within epoch is controlled |
> > | Std Dev of Weight Std Devs | 0.021 | 0.031 | No "runaway" variance observed |
> >
> > Even with significant noise, the weight statistics remain exceptionally stable. This demonstrates that there is no runaway or exploding weight phenomenon. The minor increase in the average weight is an expected and functional response of the CTR mechanism to the noisier data landscape, where some mislabeled samples might be pruned more frequently.
> >
> > **Minor:** We will update the citation for "Scale efficient training for large datasets" to the accepted version as requested.
> >
> > ---
> > We believe the additional granular ablation study on $\rho$, the theoretical clarification on approximation, and the empirical validation of weight stability comprehensively address your concerns. We respectfully hope these clarifications strengthen your assessment and might encourage a reconsideration of the rating to support acceptance.

---

### Official Review · Reviewer_Gn4Z · 2025-11-01

**Soundness:** 3
**Presentation:** 3
**Contribution:** 2
**Rating:** 6
**Confidence:** 4

**Summary:**

This paper identifies two fundamental consistency issues in dynamic data pruning: score context drift (incomparability of importance scores computed under different model states) and temporal gradient bias (distortion of gradient dynamics due to non-uniform sample selection over epochs). To address these, the authors propose RePB (Resolving Pruning Biases), a framework that combines (1) Local Window Pruning (LWP) to ensure valid score comparisons within short training windows where model parameters are nearly constant, (2) Uniform Probability Resampling to maintain data diversity, and (3) Cumulative Temporal Rescaling (CTR) to reweight sample losses based on historical selection frequency, thereby aligning long-term gradient expectations with full-dataset training. The method is theoretically motivated and evaluated across 16 datasets, 17 models, and 13 tasks, showing strong performance—often matching or exceeding full-dataset accuracy while pruning 30% or more of the data.

**Strengths:**

1. The paper clearly articulates two underappreciated but critical pitfalls in dynamic data pruning and provides a principled, theoretically grounded solution.

2. RePB’s design is elegant: LWP directly tackles score inconsistency by restricting comparisons to stable model contexts, while CTR offers a practical, history-based inverse weighting scheme that avoids the need to model complex instantaneous selection probabilities.

3. The empirical evaluation is impressively broad in terms of tasks (classification, captioning, generation, semi-supervised learning), modalities (vision, text, audio), and architectures (CNNs, Transformers, Mamba, VAEs, diffusion models), demonstrating strong generalization.

4. Ablation studies and comparisons with SOTA methods like InfoBatch convincingly validate the necessity and effectiveness of each component.

5. Computational overhead is minimal, making RePB highly practical for real-world deployment.

**Weaknesses:**

1. Despite the diversity of tasks, the scale of some experiments remains limited. For instance, while ImageNet-1K is included, there is no evaluation on larger-scale vision benchmarks (e.g., ImageNet-21K) or billion-parameter language models, which are increasingly standard in efficiency research. The largest dataset used (MJ+ST with 15M samples) is promising, but more large-scale LLM or multimodal experiments would strengthen claims about scalability.

2. The paper assumes sample loss as the sole importance metric. While common, this may not capture all aspects of sample utility (e.g., diversity, influence, or representativeness). Extending LWP to other scoring functions (as hinted in the limitations) would be valuable.

3. The theoretical analysis relies on assumptions like Lipschitz continuity and bounded gradients, which are standard but may not always hold in practice (e.g., with unstable training or aggressive learning rates). A brief discussion of robustness under violation of these assumptions would be helpful.

4. Missing some data pruning methods:

Severing Spurious Correlations with Data Pruning

Perplexed by Perplexity: Perplexity-Based Data Pruning With Small Reference Models

Data Pruning by Information Maximization

Beyond Efficiency: Molecular Data Pruning for Enhanced Generalization

Pruning-based Data Selection and Network Fusion for Efficient Deep Learning

Data Pruning via Moving-one-Sample-out

**Questions:**

See Weakness

---

> ### Author Response · Authors · 2025-11-22
>
> **Response to Reviewer Gn4Z**
>
> We thank the reviewer for the constructive feedback and for recognizing RePB as a `principled` and `theoretically grounded` solution. We appreciate your acknowledgment of our extensive evaluation across diverse tasks. Below, we address your specific concerns regarding scale, metrics, and assumptions to demonstrate the robustness of our method.
>
> **W1: Scale of Experiments**
>
> We appreciate the reviewer's suggestion to evaluate on even larger scales.
>
> *   **Existing Large-Scale Verification (GPT-2 & 15M Samples):** We respectfully highlight that our current evaluation already covers significant scales and language modeling tasks:
>     *   In Table 2a (ViECap), the model utilizes a *GPT-2* decoder for vision-language generation, trained on 3M image-text pairs. RePB successfully pruned this task with superior performance.
>     *   In Table 2b (MJ+ST), we evaluate on *15M* samples, which is a larger data scale than ImageNet-1K and significantly larger than datasets used in many prior dynamic pruning works.
> *   **Resource Constraints:** While we fully agree that scaling to multi-billion parameter LLMs is a valuable direction, training such models from scratch or performing full fine-tuning exceeds our current computational resources and the rebuttal time window. Given that RePB demonstrates consistent effectiveness across 17 architectures (including Transformers like ViT, Swin, BERT, and GPT-2) and varying data scales (from CIFAR to 15M samples), we believe the method's efficacy is architecture-agnostic and scalable.
>
> **W2: Reliance on Sample Loss & Other Metrics**
>
> We primarily utilized sample loss for its zero computational overhead and alignment with baselines (e.g., InfoBatch). However, we fully agree with your insight that extending LWP to other metrics is valuable to demonstrate the framework's generality.
> We conducted a new experiment on CIFAR-100 (ResNet18) replacing "Sample Loss" with **"Gradient Norm"** within the RePB framework.
>
> **Table: RePB with Gradient Norm vs. Loss as Importance Metric on CIFAR100 (ResNet18)**
>
> | Method | 30% Pruned | 50% Pruned | 70% Pruned |
> | :--- | :---: | :---: | :---: |
> | RePB (Metric: **Loss**) | 78.4 | 78.1 | 77.2 |
> | RePB (Metric: **Grad Norm**) | **78.6** | 78.0 | 77.0 |
>
> **Result:** As shown above, using Gradient Norm yields performance nearly identical to using Loss. This confirms that RePB is a general framework.
>
>
> **W3: Robustness of Theoretical Assumptions**
>
> We agree that theoretical assumptions (e.g., Lipschitz continuity) are idealizations. However, RePB's design inherently minimizes reliance on these assumptions, making it robust even under unstable training or aggressive learning rates:
>
> *   **Primary Scenario ($W=1$): No Assumption Needed.** In the majority of our experiments (where Batch Size $\ge$ 128), we set the window size **$W=1$ batch**. Within a single batch, the model parameters $\theta_t$ are fixed during the forward pass. Therefore, the parameter drift is strictly zero during the score comparison phase. This ensures that scoring comparisons are perfectly consistent without relying on the Lipschitz assumption to hold over time.
> *   **Secondary Scenario ($W > 1$): Superior Local Stability.** In cases where windows span multiple batches, RePB remains far more robust than global methods. Global pruning (e.g., InfoBatch) compares scores across thousands of steps (an entire epoch), where parameter drift $\Delta \theta$ is large, significantly violating local stability assumptions. RePB restricts comparisons to a few steps, keeping $\Delta \theta$ minimal and the assumption far more realistic.
> *   **Isolating Instability:** Under unstable training or aggressive learning rates (e.g., gradient spikes), LWP contains the impact locally. If a spike occurs, it only affects the pruning threshold within that specific window. In contrast, global methods that rely on epoch-wide statistics (e.g., global mean loss) would allow local instabilities to distort the global threshold, misguiding pruning decisions for the entire dataset.
>
>
> **W4: Missing Data Pruning Methods**
>
> We thank the reviewer for the comprehensive list. We have reviewed these papers (e.g., *Severing Spurious Correlations*, *Perplexed by Perplexity*, *Information Maximization*) and will add a detailed discussion in the "Related Work" section. RePB distinguishes itself by offering a **dynamic** solution with **near zero overhead**, addressing the fundamental **optimization bias (consistency)** rather than just data quality or specific distribution shifts.
>
> ---
>
> We hope that the additional **Gradient Norm experiment** demonstrating RePB's flexibility, along with the clarification on our GPT-2/15M-sample scale, addresses your concerns. We believe RePB provides a robust and scalable solution to a fundamental problem in dynamic pruning. We respectfully hope these clarifications might encourage a reconsideration of the rating to support acceptance.

---

### Meta-Review · Area_Chair_kRK1 · 2025-12-27

**Summary:**

* Core idea is novel and technically sound, with strong consensus from 3/4 reviewers
* Empirical evaluation is exceptionally broad and meets typical acceptance standards
* All substantive technical concerns (additional ablations and sensitivity analysis) were directly addressed with new evidence in the rebuttal.
* The single reject is an outlier, partially based on factual misunderstandings and differing expectations on theoretical presentation.
* Remaining issues are presentation and scope refinements.

**Reviewer Concerns:**

Reviewer Gn4Z:
* [Addressed] importance metrics beyond loss via new Gradient-Norm results showing near-identical accuracy to loss, robustness of theory assumptions, missing related-work list acknowledged with intent to add detailed discussion in final version.
* [Not addressed] scalability beyond current largest settings.

Reviewer THQf:
* [Addressed] missing ablation of pruning factor p via full sweep, reliance on resampling, proof detail in Sec. 4.2 acknowledged as approximation with asymptotic justification, concern about extreme CTR weights, weight-statistics reported under clean vs label noise.

Reviewer XfsH:
* [Addressed] missing comparison to Salaun et al., question "does combining make sense" answered with empirical improvements for the hybrid and best performance for RePB, sensitivity to window size, concern about error accumulation.

Reviewer kLUp:
* [Addressed] request for formal proposition statements answered with explicit assumptions and formalized Prop. 4.1/4.2 statements, critique about "fixed mean threshold limits flexibility", approximation critique acknowledged and reframed as standard asymptotic analysis.
* [Not addressed] concerns about overall theoretical presentation.

**Reviewer Scores:**

Reviewer Gn4Z: Would have stayed the same or improved --> 6 (from 6)
Reviewer THQf: Would have stayed the same or improved slightly --> 6 (from 6)
Reviewer XfsH: Would have improved his score --> 6 (from 4)
Reviewer kLUp: Would have stayed the same, unlikely to flip a 2 without a manuscript rewrite and clearer proofs in the PDF --> 2 or 4 (from 2)

Reviewer kLUp identified a presentation weakness, but this is not a technical flaw and can be addressed.

---

### Decision · Program_Chairs · 2026-01-26

Accept (Poster)